# Literacy in the Scope of Radiation Protection for Healthcare Professionals Exposed to Ionizing Radiation: A Systematic Review

**DOI:** 10.3390/healthcare12202033

**Published:** 2024-10-12

**Authors:** Belinda V. Rodrigues, Pedro C. Lopes, Anna C. Mello-Moura, Javier Flores-Fraile, Nelio Veiga

**Affiliations:** 1Department of Surgery, Universidad de Salamanca, 37008 Salamanca, Spain; belindavicente@hotmail.com (B.V.R.); j.flores@usal.es (J.F.-F.); 2Center for Interdisciplinary Research in Health, Faculty of Dental Medicine, Universidade Católica Portuguesa, 3504-505 Viseu, Portugal; acmoura@ucp.pt (A.C.M.-M.); nveiga@ucp.pt (N.V.)

**Keywords:** radiation protection, healthcare professionals, literacy, knowledge

## Abstract

**Background**: The use of radiation is important in different medical procedures, and to ensure a high level of good medical practice, radiation protection (RP) should be seen as a very important subject. This review shows information about the knowledge in the scope of radiation protection among healthcare professionals exposed to ionizing radiation. There are not many studies that evaluate the radiation exposure of healthcare professionals. **Methods**: A systematic search was performed, using PRISMA guidelines, in Pubmed and Scopus databases and manually to identify relevant articles to answer the PICOS question, “Is there an adequate level of literacy in the scope of radiation protection among healthcare professionals exposed to ionizing radiation?”. This systematic review included cross-sectional studies with the following inclusion criteria: (i) in Portuguese, Spanish, or English; (ii) about literacy in the scope of radiation protection; (iii) published between 2017 and 2024; and (iv) participants must be dentists, radiographers, doctors, and nurses. The JBI critical assessment tool was used to assess the risk of bias. **Results**: The search identified 566 potentially relevant references, which, after applying inclusion/exclusion criteria, resulted in 12 articles. Studies found that the overall knowledge of these healthcare workers was unsatisfactory, and a lack of knowledge in radiation protection negatively affects health services’ quality. Training is essential and must emphasize how radiation exposure can be minimized, safeguarding health professionals’ trust and sense of security. Results showed that more years of experience make workers more attentive to protection measures, suggesting that training strategies focused on basic radiological risks and radiation safety are needed. **Conclusions**: Key findings recommend implementing a standardized national training program on the basic principles and safety of ionizing radiation for all healthcare professionals.

## 1. Introduction

Recently, the International Association for Radiological Protection (IARP) came up with a “RP Safety Culture” concept as a “combination of knowledge, values, behaviors, and experiences of RP in all its aspects for patients, workers, population, or environment, and in all exposure situations, combining scientific and social dimensions” [1].

The range of uses of ionizing radiation in the medical field is wide. In 2008, throughout the whole world, there were nearly one million radiation therapy procedures, 3.6 billion diagnostic and interventional radiological procedures, which included dentistry, and 30 million nuclear medicine procedures [1]. The number of procedures has risen in recent years due to the growing frequency of operations under fluoroscopic guidance [2]. Although IR is used in the medical field, it can be a substantial public health problem because it can cause harm. To guarantee the equilibrium between using IR in medical procedures and minimizing the risks of radiation effects to the patient that healthcare workers and the public can obtain, a systematic approach is a good option [3].

In diagnostics, interventional procedures, and treatments, the use of IR is wide. To certify the good medical practices are being used, a larger system should be created to guarantee RP [4]. All safety guides should include guidance to make sure RP and safety of radiation sources concerning patients, workers, caregivers, comforters, volunteers, research workers, and the public while using IR for medical purposes. It includes radiological procedures in diagnostic radiology (including dentistry), image guidance during interventional procedures, nuclear medicine, and radiation therapy. The need for image guidance in radiological procedures and X-ray diagnostics can be performed by various medical specialties, such as cardiology, vascular surgery, urology, orthopedics, gastroenterology, gynecology, and anesthetics [4]. Currently, the IARP suggests the concept of a “RP Safety Culture” that may be capable of being outlined as [5] “the combination of knowledge, values, behaviors, and experiences of RP in all its aspects for patients, workers, population, or environment, and in all exposure situations, combining scientific and social dimensions” [6].

Pediatric patients are a very special population when it comes to X-ray exposure. Although pediatric anesthesiologists are aware of it, usually they are not sensible in the sense of decreasing the intensity of the exposure. Usually, because at their workplace the RP Safety Culture does not exist [7]. Routinely, there are various health professions that use IR to perform procedures, such as in the diagnostic, interventional, and treatment fields. The success of the RP measures depends on the knowledge and expertise of the healthcare worker involved. In each case, the degree of engagement of the patient in all decisions is a very important factor [4]. Despite the abundance of epidemiological and biological research investigations, the low effects on health resulting from low exposure to IR are inconsistent. There is a gap in understanding experimental and epidemiological studies because studies are not stable [8,9]. Due to radiation exposure, healthcare workers are exposed to different radio wave levels. They can be related to acute complications such as dermatitis, mucositis, and hair loss, but they can be related to long-term complications too. The long-term complications that cause damage to the normal DNA function include cataracts, skin problems, genetic problems, and cancer [10,11]. In order to highlight the promotion of safety, it has a very important subject related to RP in the medical field. It is extremely important to involve professionals in education, training, qualifications, and competence of the respective profession [12]. To guarantee the correct and necessary demands for education, training, qualifications, and competence in RP, there is a need for a regulatory body. The solicitation should be at the same time as the application asks for authorization (to the regulatory body) and during periodic inspection of the medical radiation facility [13]. The most important key to radiographers and radiologists’ performance is to offer the best health benefit to their patients while always keeping their protection [14].

An ideal approach to summarize literature related to a certain social or healthcare topic is a systematic review. A systematic review of evidence is a well-accepted method committed to being ideal; it can furnish a summary of the literature to notify decision makers [15]. Findings indicate that healthcare personnel working with IR sources have a problem transforming their knowledge and attitudes about radiation safety into behavior [16]. The aim of this review is to clarify information about the knowledge in the scope of radiation protection among healthcare professionals exposed to ionizing radiation, especially because there are not many studies that evaluate the radiation exposure of healthcare professionals [17].

## 2. Materials and Methods

This systematic review was conducted following the Preferred Reporting Items for Systematic reviews and Meta-Analysis (PRISMA) guidelines [18] and has been recorded in OSF Registries. The OSF database with the registration DOI can be found on the following link: http://osf.io/fmvua/ (accessed on 24 August 2024).

To fulfill the proposed objective, a PICO(S) question was formulated. “Is there an adequate level of literacy in the scope of radiation protection among healthcare professionals exposed to ionizing radiation?”. Below is the defined PICO(S) question:P (Participants): healthcare professionals;I/E (Intervention/Exposure): literacy;C (Comparison): in the scope of radiation protection;O (Outcome): exposed to ionizing radiation;S (Study type): systematic review.

The search strategies were based on the PICO(S) question, developed for the PubMed database and adapted for the Scopus and Cochrane databases, to identify relevant articles from 2017 to the end of 2024. The results from the different bases were cross-referenced to locate and eliminate the duplicates. The complete search strategy for PubMed is as follows:(healthcare providers) AND (radiation protection) AND (literacy OR knowledge)For the Scopus database, the following strategy was used:(healthcare AND providers) AND (radiation AND protection) AND (literacy OR knowledge) AND PUBYEAR > 2017 AND PUBYEAR < 2025 AND (LIMIT-TO (LANGUAGE, “Spanish”) OR LIMIT-TO (LANGUAGE, “English”) OR LIMIT-TO (LANGUAGE, “Portuguese”)).

This systematic review included cross-sectional and cohort studies, complying with the following inclusion criteria: (i) being in Portuguese, Spanish, or English; (ii) being about literacy in the scope of radiation protection; (iii) having been published between 2017 and 2024; (iv) participants must be dentists, radiographers, doctors (orthopedists, neurosurgeons, anesthesiologists, and cardiologists), and nurses; and (v) full text available. In this review, we excluded systematic reviews, case reports, expert opinions, and studies with less than 40 participants.

## 3. Results

After the literature search, two independent researchers (BVR and PCL) filtered relevant articles that fit this study by analyzing their titles and abstracts for study selection. Any disagreements between the reviewers were discussed with a third author (NV). Cohen’s Kappa test was performed to assess reviewers’ agreement. Rayyan’s Intelligent Systematic Review Platform was used to assist in the systematic review process [19]. The study selection process is shown in Figure 1.

Reviewers extracted data independently from the articles selected for analysis. The information collected during data extraction included the title of the article, year of publication, authors, study design, type of participants, number of participants, and the evaluated outcome. Results are shown in Table 1.

Table 1 shows the primary characteristics of the 12 studies selected for this systematic review. All studies selected are surveys. Most of the studies reported poor knowledge about radiation safety [19,20,21,22,23,24,26,27,28,29,30]. In all studies, radiation protection equipment exists, but not all professionals use it [23,25].

The attitude and behavior towards ionizing radiation are limited and less accurate in professionals with fewer years of experience, referring to a lack of training towards radiation protection [22,23,25,26]. The knowledge of regulatory body legislation about radiation safety was referred to by [20,25,28] as not satisfactory. Dosimetry is a subject mentioned by two studies, one of which mentioned satisfactory knowledge [19] and the other that half of the professionals use it [25].

To evaluate the methodological quality of the studies (Table 2), the critical assessment tool—the Joanna Briggs Institute (JBI)—was used [32]. Cross-sectional studies (surveys) were analyzed regarding the quality of the study according to the JBI criteria, and the results of the analysis are presented in Table 2. All aspects of the analysis were approached except for the ones regarding confounding factors. In some articles, this aspect was not identified, and strategies to deal with confounding factors were not always stated.

## 4. Discussion

The principle of this review is to integrate the information available within the scope of radiological protection for healthcare professionals exposed to ionizing radiation. Goula et al. (2021) [20] conducted a survey to assess the level of knowledge about radiation protection safety among healthcare professionals. The present review included 132 nurses, radiographers, and medical doctors from various specialties working in operating rooms where ionizing radiation is used. This study found that the overall knowledge of these healthcare workers was unsatisfactory. Results suggested that ongoing training is essential to address this issue. Such training should focus on minimizing radiation exposure, thereby enhancing health professionals’ confidence and comfort; it should considerably improve their working conditions. The findings are consistent with other studies in literature, such as De Gruyter et al. (2021) [33]. In general, these studies explain the limited knowledge about radiation safety, poor training, and scarce usage of radiation protective equipment. Raza et al. (2021) [13] conducted a national inquiry to evaluate the level of basic principles of radiation and legislation. The sample included 406 orthopedic surgeons working in operating rooms where ionizing radiation is used. This study revealed a lack of basic, specialized, and adequate training concerning protection safety, which negatively impacts the provision of health services. The findings are consistent with other studies in literature, such as Ihle et al. (2018) [24].

It is necessary to adapt the training program, especially regarding the radiological risks and radiation protective safety issues. Yurt et al. (2022) [22] conducted a survey with a sample that included 66 dentists who work with ionizing radiation. Continuing educational and adequate training in agreement with the standards set by national authorities are of great importance and should be encouraged. A succession of studies appears to confirm the findings of this survey in the literature. For example, Park et al. (2021) [23] conducted a cross-sectional study to evaluate the level of knowledge about radiation protection safety among healthcare professionals. The sample included 129 nurses working in operating rooms where ionizing radiation is used. This study revealed that healthcare workers lacked sufficient knowledge in this area. The findings indicated that the insufficient knowledge of health professionals in radiation protection led to a decrease in the quality of health services. They suggested that ongoing training is essential to address this issue. Similar findings were reported in the literature by Jeyasugiththan et al. (2023) [34].

Falavigna et al. (2018) [25] conducted a survey with 371 spine surgeons working in operating rooms where ionizing radiation is used. This study found that the lack of knowledge and measures taken were not sufficient. Standards set by national authorities have great importance. A series of studies confirmed the findings of this article, such as Sain et al. (2023) [28], which highlighted the need for education and adequate training in agreement with national standards set by national authority has a great importance and should be encouraged. Uthirapathy et al. (2022) [29] conducted a survey with 185 cardiologists that came to reinforcing the need to adapt training and continuing education in conformity with the standards set by national standards. Snowden et al. (2022) [30] conducted a survey in Scotland with a sample including 72 orthopedic surgeons working in operating rooms where ionizing radiation is used, with similar findings. Early radiation training was seen as a way to improve practice and reduce radiation exposure to staff during the interventions in the operating room. Radiation protection safety practices need to increase.

The importance of adequate training strategies to meet standards set by national legislation was demonstrated in Bernelli et al. (2024) [31].

An et al. (2018) [26] conducted a survey with a sample that included 207 dentists who use ionizing radiation. It showed that there was a poor understanding of radiation safety among those professionals.

The author concluded that the provision of quality health services can be compromised by the deficiency in elementary and specialized knowledge regarding RPS by the healthcare workers. The refreshing coaching of the staff appeared to be the best solution to improve good practices.

In order to safeguard health professionals trust and comfort, the teaching program must emphasize how radiation exposure can be minimized; it will upgrade their working environment.

Macía-Suárez et al. (2018) [27] conducted a survey with 63 radiologists working in operating rooms where ionizing radiation is used. This study found that healthcare workers had a significant knowledge gap in radiation safety. Both studies revealed that seniority, or more years of experience, would make workers more attentive to radiation protection measures, suggesting there is a need to adapt the type of training strategies.

The perspective of previous studies and the different working hypotheses should be considered in author discussion of the results. The findings and their implications should be discussed in a wider context. Other study directions may also be considered.

## 5. Conclusions

All healthcare professionals in the reviewed studies demonstrate a low level of literacy and awareness regarding radiation protection. However, it emphasizes the importance of continuous education and adequate training on RP principles and safety, standardized at the national level for all healthcare professions.

Key findings recommend implementing a standardized national training program on the basic principles and safety of ionizing radiation for healthcare professionals, tailored to each profession. To assess the usefulness of the continuing education input on the safety culture and identify any remaining gaps (around radiation protection), it is recommended to have more studies.

## Figures and Tables

**Figure 1 healthcare-12-02033-f001:**
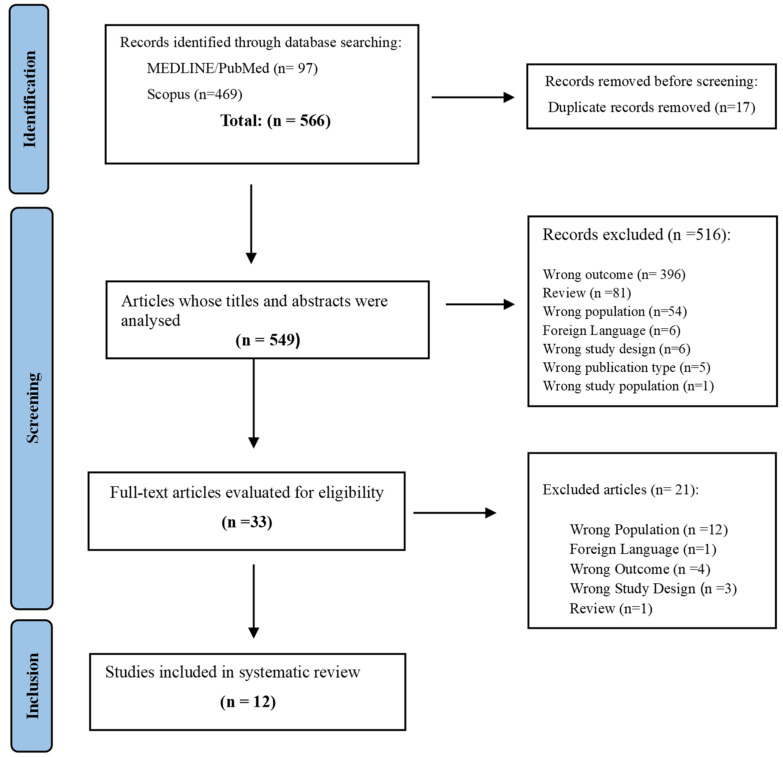
Overview of article selection procedure according to PRISMA guidelines.

**Table 1 healthcare-12-02033-t001:** Attributes of the 12 studies selected for this review.

Authors Year	Study Design	Sample Size	Types of Participants	Main Results	Conclusions
Goula et al. 2021 [20]	Survey	132	Nurses Radiographers Doctors	Elementary education of K. R. P. S.—not satisfactory Radiation Protection Safety Equipment—healthcare professionals experience discomfort Negativism towards R. P. S. E.—healthcare professionals tended to have it Knowledge of dosimetry—was satisfactory Misunderstanding about R. and R. P.—healthcare professionals had some Influence of number of years of experience—NR Knowledge of Regulatory Body Legislation about Radiation Protection—NR Training in Ionizing Radiation Safety—NR Attitude and behavior towards Ionizing Radiation—NR	Health professionals concerning protection safety have a low level of adequate and specialized training. The national authority standards emphasize the relevance of continuing education and adequate training.
Raza et al. 2021 [21]	Survey	406	Orthopedics Surgeons	B. K. R. P. S.—was limited Radiation Protection Safety Equipment—healthcare professionals had a poor uptake Negativism towards R. P. S. E.—healthcare professionals tended to have it Knowledge of dosimetry: NR Misunderstanding about R. and R. P.—healthcare professionals had Influence of number of years of experience—NR Knowledge of Regulatory Body Legislation about Radiation Protection—was limited Training in Ionizing Radiation Safety—only 19% had adequate training Attitude and behavior towards Ionizing Radiation—NR	Regarding protection and safety, many orthopedic surgeons in the UK have an elementary training. This reflects on the good medical practices.
Yurt et al. 2022 [22]	Survey	66	Dentists	B. K. R. P. S.—moderate Radiation Protection Safety Equipment: moderate Negativism towards R. P. S. E.—NR Knowledge of dosimetry—NR Misunderstanding about R. and R. P.—NR Influence of number of years of experience—decline perception by age and years of experience Knowledge of Regulatory Body Legislation about Radiation Protection—NR Training in Ionizing Radiation Safety—NR Attitude and behavior towards Ionizing Radiation—satisfactory	Regarding protection safety, dentists have an elementary training. The national authority standards emphasize the relevance of continuing education and adequate training.
Park et al. 2021 [23]	Survey	129	Nurses	B. K. R. P. S.—average Radiation Protection Safety Equipment: 62.6% reported wearing protective equipment Negativism towards R. P. S. E.—NR Knowledge of dosimetry—NR Misunderstanding about R. and R. P.—NR Influence of number of years of experience—NR Knowledge of Regulatory Body Legislation about Radiation Protection—NR Training in Ionizing Radiation Safety—54.5% showed no expertise on RP education Attitude and behavior towards Ionizing Radiation—minority	Regarding protection safety, nurses have an elementary training. The national authority standards emphasize the relevance of continuing education and adequate training.
Ihle et al. 2018 [24]	Survey	63	Dentists	B. K. R. P. S.—inconsistent Radiation Protection Safety Equipment—NR Negativism towards R. P. S. E.—NR Knowledge of dosimetry—NR Misunderstanding about R. and R. P.—need for spreading awareness (69.8%) Influence of number of years of experience—NR Knowledge of Regulatory Body Legislation about Radiation Protection—80.3% follow the Australian Radiation Protection and Nuclear Safety Agency guidelines Training in Ionizing Radiation Safety—38.1% had undertaken an RP course or training in the past 2 years Attitude and behavior towards Ionizing Radiation—need for spreading awareness (69.8%)	Regarding protection safety, dentists have an elementary training.
Falavigna et al. 2018 [25]	Survey	371	Spine Surgeons	B. K. R. P. S.—not satisfactory Radiation Protection Safety Equipment: 64.2% use an ear and thyroid protector Negativism towards R. P. S. E.—NR Knowledge of dosimetry—75.7% never or rarely used Misunderstanding about R. and R. P.—NR Influence of number of years of experience—NR Knowledge of Regulatory Body Legislation about Radiation Protection—NR Training in Ionizing Radiation Safety—NR Attitude and behavior towards Ionizing Radiation—NR	The national authority standards emphasize the relevance of continuing education and adequate training to minimize intraoperative radiation exposure.
An et al. 2018 [26]	Survey	207	Dentists	B. K. R. P. S.—mot satisfactory Radiation Protection Safety Equipment—NR Negativism towards R. P. S. E—NR Knowledge of dosimetry—57% use a dosimeter badge Misunderstanding about R. and R. P.—NR Influence of number of years of experience –dentists with a decade of experience were more careful with RP procedures Knowledge of Regulatory Body Legislation about Radiation Protection—not satisfactory Training in Ionizing Radiation Safety—83% participated in a radiation safety program Attitude and behavior towards Ionizing Radiation—professionals with a decade of experience or more seemed to be more cautious	Dentists with more than 10 years of experience were more careful about radiation protection measures. Regarding protection safety, dentists have an elementary training.
Macía-Suárez et al. 2018 [27]	Survey	63	Radiologists	B. K. R. P. S.—low Radiation Protection Safety Equipment: most use it sometimes Negativism towards R. P. S. E—NR Knowledge of dosimetry—NR Misunderstanding about R. and R. P.—NR Influence of number of years of experience—level becomes higher with the higher number of years of experience Knowledge of Regulatory Body Legislation about Radiation Protection—NR Training in Ionizing Radiation Safety—NR Attitude and behavior towards Ionizing Radiation—poor uptake	Regarding protection safety, radiologists have an elementary training.
Sain et al. 2023 [28]	Survey	49	Orthopedic Surgeons Anesthetists Radiographers Theater Nurses Other Healthcare Professionals	B. K. R. P. S.—theatre radiographers were the only one fully aware Radiation Protection Safety Equipment—46% use protective lead apron Negativism towards R. P. S. E—NR Knowledge of dosimetry—NR Misunderstanding about R. and R. P.—NR Influence of number of years of experience—NR Knowledge of Regulatory Body Legislation about Radiation Protection—NR Training in Ionizing Radiation Safety—less than half had formal training Attitude and behavior towards Ionizing Radiation—poor	The findings from the audit highlight: The national authority standards emphasize the relevance of continuing education and adequate training.
Uthirapathy et al. 2022 [29]	Survey	185	Cardiologists	B. K. R. P. S.—poor Radiation Protection Safety Equipment—87–100% used a lead apron Negativism towards R. P. S. E—NR Knowledge of dosimetry—40% used it regularly Misunderstanding about R. and R. P.—NR Influence of number of years of experience—NR Knowledge of Regulatory Body Legislation about Radiation Protection—poor uptake Training in Ionizing Radiation Safety—NR Attitude and behavior towards Ionizing Radiation—was limited	Cardiologists were aware of radiation safety measures. The national authority standards emphasize the relevance of continuing education and adequate training.
Snowden et al. 2022 [30]	Survey	72	Orthopedic Surgeons	B. K. R. P. S.—poor Radiation Protection Safety Equipment—always wear a lead apron Negativism towards R. P. S. E—NR Knowledge of dosimetry—most do not have it Misunderstanding about R. and R. P.—NR Influence of number of years of experience—years of experience increase knowledge Knowledge of Regulatory Body Legislation about Radiation Protection—NR Training in Ionizing Radiation Safety—51.4% had teaching/education experience on radiation safety Attitude and behavior towards Ionizing Radiation—they know the importance of radiation protection	Early formal radiation training would like to improve practice and reduce radiation exposure on staff. The national authority standards emphasize the relevance of continuing education and adequate training.
Bernelli et al. 2024 [31]	Survey	237	Physicians Nurses Radiology Technologists	B. K. R. P. S.—poor Radiation Protection Safety Equipment—they should have a dedicated lead apron Negativism towards R. P. S. E—positive impression about being regularly checked Knowledge of dosimetry—NR Misunderstanding about R. and R. P.—NR Influence of number of years of experience—NR Knowledge of Regulatory Body Legislation about Radiation Protection—non-physicians were more aware Training in Ionizing Radiation Safety—has insufficient programs and requires more support training to maintain skills Attitude and behavior towards Ionizing Radiation—should be more adequate	The national authority standards emphasize the relevance of continuing education and adequate training. This will be relevant to ensuring equality in career opportunities for all genders.

Legend: B. K. R. P. S.—Basic Knowledge Radiation Protection Safety; Negativism towards R. P. S. E—Negativism towards Radiation Protection Safety Equipment; Misunderstanding about R. and R. P.—Misunderstanding about Radiation and Radiation Protection.

**Table 2 healthcare-12-02033-t002:** Risk of bias according to JBI, the critical assessment tool, to cross-sectional studies.

Authors Year	Were the Criteria for Inclusion in the Sample Clearly Defined?	Were the Study Subjects and the Settings Described in Detail?	Was the Exposure Measured in a Valid and Reliable Way?	Were Objective, Standard Criteria Used for Measurement?	Were Confounding Factors Identified?	Were Strategies to Deal with Confounding Factors Stated?	Were the Outcomes Measured in a Valid and Reliable Way?
Goula et al., 2021 [1]	yes	yes	yes	yes	In part	In part	yes
Raza et al., 2021 [2]	yes	yes	yes	yes	no	no	yes
Yurt et al., 2022 [3]	yes	yes	yes	yes	In part	In part	yes
Park et al., 2021 [4]	yes	yes	yes	yes	In part	In part	yes
Ihle et al., 2018 [5]	yes	yes	yes	yes	no	yes	yes
Falavigna et al., 2018 [6]	yes	yes	yes	yes	no	yes	yes
An et al., 2018 [7]	yes	yes	yes	yes	In part	yes	yes
Macía-Suárez et al., 2018 [8]	yes	yes	yes	yes	yes	yes	yes
Sain et al., 2023 [9]	In part	yes	yes	yes	yes	yes	yes
Uthirapathy et al., 2022 [10]	yes	yes	yes	yes	yes	yes	yes
Snowden et al., 2022 [11]	In part	yes	yes	yes	In part	In part	yes
Bernelli et al., 2024 [12]	In part	yes	yes	yes	yes	yes	yes

## Data Availability

No new data were created or analyzed in this study.

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
