# Peer review of "Literacy in the Scope of Radiation Protection for Healthcare Professionals Exposed to Ionizing Radiation: A Systematic Review"

_healthcare, 2024, doi:10.3390/healthcare12202033_

Round 1
Reviewer 1 Report
Comments and Suggestions for Authors
Refer to the attached file

Author Response
We attach the pdf with the comments and suggestions made by Reviewer 1.

Reviewer 2 Report
Comments and Suggestions for Authors
Dear Authors,
This study is significant and interesting for readers.
Please consider the following points to improve this paper.
1. Line 90 : Please describe why the place of this study is limited to Portugal? Does this review include papers reporting cases in other countries than Portugal?
2. Line 100 : Since almost all papers included in this review were not intervention study. Therefore, authors can replace I (intervention) to E(exposure).
3. Please describe why the term during which papers published were limited later than 2017.
4. Targets of studies included this review were healthcare professionals, so I was surprised that even in professionals there were wide lack of knowledge and literacy. Around the world, even publics may have a chance of exposure of radiation due to disaster or accident. How do authors think whether the results of this study is applicable to radiation protection for publics?
Author Response
Thank you for the comments made. We send the answers to each one of the questions made:
- Line 90: Please describe why the place of this study is limited to Portugal? Does this review include papers reporting cases in other countries than Portugal?
Answer: It was a mistake in the first version. I have removed “Portugal”.
- Line 100: Since almost all papers included in this review were not intervention study. Therefore, authors can replace I (intervention) to E(exposure).
Answer: The correction has been made in the manuscript.
- Please describe why the term during which papers published were limited later than 2017.
Answer: The authors defined that the analysis should be done by the selection and description of the scientific articles in the last 7 years. We verified that this field of radiology protection appeared more in 2017 and has been in development ever since.
- Targets of studies included this review were healthcare professionals, so I was surprised that even in professionals there were wide lack of knowledge and literacy. Around the world, even publics may have a chance of exposure of radiation due to disaster or accident. How do authors think whether the results of this study is applicable to radiation protection for publics?
Answer: The results of this study will permit the implementation and standardization of national training program on the basic principles and safety of ionizing radiation for healthcare professionals, tailored to each profession.
Round 2
Reviewer 1 Report
Comments and Suggestions for Authors
There is no novelty on this paper.
Author Response
Thank you again for the opportunity of preparing a second round of revisions. The whole manuscript was re-read and information was improved and reorganized to improve coherence and logic in the presentation of the information.